# Demystifying the Adversarial Robustness of Random Transformation Defenses

## Chawin Sitawarin, Zachary Golan-Strieb, David Wagner

EECS Department, University of California, Berkeley
chawins@berkeley.edu, zacharyjgs@berkeley.edu, daw@cs.berkeley.edu

## Abstract

Current machine learning models suffer from evasion attacks (i.e., adversarial examples) raising concerns in security-sensitive settings such as autonomous vehicles. While many countermeasures may look promising, only a few withstand rigorous evaluation. Recently, defenses using random transformations (RT) have shown impressive results, particularly BaRT (Raff et al. 2019) on ImageNet. However, this type of defense has not been rigorously evaluated, leaving its robustness properties poorly understood. The stochasticity of these models also makes evaluation more challenging and many proposed attacks on deterministic models inapplicable. First, we show that the BPDA attack (Athalye, Carlini, and Wagner 2018) used in BaRT's evaluation is ineffective and likely overestimates its robustness. We then attempt to construct the strongest possible RT defense through the informed selection of transformations and Bayesian optimization for tuning their parameters. Furthermore, we create the strongest possible attack to evaluate our RT defense. Our new attack vastly outperforms the baseline, reducing the accuracy by 83% compared to the 19% reduction by the commonly used EoT attack ($4.3\times$ improvement). Our result indicates that the RT defense on Imagenette dataset (ten-class subset of ImageNet) is not robust against adversarial examples. Extending the study further, we use our new attack to adversarially train RT defense (called AdvRT). However, the attack is still not sufficiently strong, and thus, the AdvRT model is no more robust than its RT counterpart. In the process of formulating our defense and attack, we perform several ablation studies and uncover insights that we hope will broadly benefit scientific communities studying stochastic neural networks and their robustness properties.

## 1   Introduction

Today, deep neural networks are widely deployed in safety-critical settings such as autonomous driving and cybersecurity. Despite their effectiveness at solving a wide-range of challenging problems, they are known to have a major vulnerability. Tiny crafted perturbations added to inputs (so called *adversarial examples*) can arbitrarily manipulate the outputs of these large models, posing a threat to the safety and privacy of the millions of people who rely on existing ML systems. The importance of this problem has drawn substantial atten-

tion, and yet we have not devised a concrete countermeasure as a research community.

Adversarial training (Madry et al. 2018) has been the foremost approach for defending against adversarial examples. While adversarial training provides increased robustness, it results in a loss of accuracy on benign inputs. Recently, a promising line of defenses against adversarial examples has emerged. These defenses randomize either the model parameters or the inputs themselves (Lecuyer et al. 2019; He, Rakin, and Fan 2019; Raff et al. 2019; Liu et al. 2019; Xie et al. 2018; Zhang and Liang 2019; Bender et al. 2020; Liu et al. 2018; Cohen, Rosenfeld, and Kolter 2019; Dhillon et al. 2018; Guo et al. 2018). Introducing randomness into the model can be thought of as a form of smoothing that removes sinuous portions of the decision boundary where adversarial examples frequently lie (He, Li, and Song 2018). Among these randomization approaches, Raff et al. (2019) propose Barrage of Random Transforms (BaRT), a new defense which applies a large set of random image transformations to classifier inputs. They report a $24\times$ increase in robust accuracy over previously proposed defenses.

Despite these promising results, researchers still lack a clear understanding of how to properly evaluate random defenses. This is concerning as a defense can falsely appear more robust than it actually is when evaluated using sub-optimal attacks (Athalye, Carlini, and Wagner 2018; Tramer et al. 2020). Therefore, in this work, we improve existing attacks on randomized defenses, and use them to rigorously evaluate BaRT and more generally, random transformation (RT) defenses. We find that sub-optimal attacks have led to an overly optimistic view of these RT defenses. Notably, we show that even our best RT defense is much less secure than previously thought, formulating a new attack that reduces its security (from 70% adversarial accuracy found by the baseline attack to only 6% on Imagenette).

We also take the investigation further and combine RT defense with adversarial training. Nevertheless, this turns out to be ineffective as the attack is not sufficiently strong and only generates weak adversarial examples for the model to train with. The outcomes appear more promising for CIFAR-10, but it still lacks behind deterministic defense such as Madry et al. (2018) and Zhang et al. (2019). We believe that stronger and more efficient attacks on RT-based models will be necessary not only for accurate evaluation of the

stochastic defenses but also for improving the effectiveness of adversarial training for such models.

To summarize, we make the following contributions:

- We show that non-differentiable transforms impede optimization during an attack and even an adaptive technique for circumventing non-differentiability (i.e., BPDA (Athalye, Carlini, and Wagner 2018)) is not sufficiently effective. This reveals that existing RT defenses are likely non-robust.
- To this end, we suggest that an RT defense should only use differentiable transformations for reliable evaluations and compatibility with adversarial training.
- We propose a new state-of-the-art attack for RT defense that improves over EoT (Athalye et al. 2018) in terms of both the loss function and the optimizer. We explain the success of our attack through the variance of the gradients.
- Improve the RT scheme by using Bayesian optimization for hyperparameter tuning and combining it with adversarial training which uses our new attack method instead of the baseline EoT.

## 2 Background and Related Works

### 2.1 Adversarial Examples

Adversarial examples are carefully perturbed inputs designed to fool a machine learning model (Szegedy et al. 2014; Biggio et al. 2013; Goodfellow, Shlens, and Szegedy 2015). An adversarial perturbation $\delta$ is typically constrained to be within some $\ell_p$-norm ball with a radius of $\epsilon$. The $\ell_p$-norm ball is a proxy to the "imperceptibility" of $\delta$ and can be thought of as the adversary's budget. In this work, we primarily use $p = \infty$ and only consider adaptive white-box adversary. Finding the worst-case perturbation $\delta^*$ requires solving the following optimization problem:

$$x_{\text{adv}} = x + \delta^* = x + \arg\max_{\delta : \|\delta\|_p \leq \epsilon} L(x + \delta, y) \qquad (1)$$

where $L : \mathbb{R}^d \times \mathbb{R}^C \to \mathbb{R}$ is the loss function of the target model which, in our case, is a classifier which makes predictions among $C$ classes. Projected gradient descent (PGD) is often used to solve the optimization problem in Eqn. 1.

### 2.2 Randomization Defenses

A number of recent papers have proposed defenses against adversarial examples which utilize inference-time randomization. One common approach is to sample weights of the network from some probability distribution (Liu et al. 2018; He, Rakin, and Fan 2019; Liu et al. 2019; Bender et al. 2020). In this paper, we instead focus on defenses that apply random transforms to the input (Raff et al. 2019; Xie et al. 2018; Zhang and Liang 2019; Cohen, Rosenfeld, and Kolter 2019), many of which claim to achieve state-of-the-art robustness. Unlike prior evaluations, we test these defenses using a wide range of white-box attacks as well as a novel stronger attack. A key issue when evaluating these schemes is that PGD attacks require gradients through the entire model pipeline, but many defenses use non-differentiable transforms. As we show later, this can cause evaluation results to be misleading.

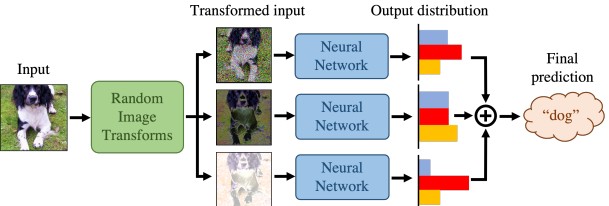

Figure 1: An illustration of a random transformation (RT) defense against adversarial examples. Transformations of different types and parameters are sampled and applied sequentially to multiple copies of the input. All of the transformed inputs are then passed to a single neural network, and the outputs are combined to make the final prediction.

Different works have tried applying different random transformations to their inputs. Xie et al. randomly resize and pad images (Xie et al. 2018). While this defense ranked second in the NeurIPS 2017 adversarial robustness competition, their security evaluation did not consider adaptive attacks where the adversary has full knowledge of the transformations.

Zhang et al. (Zhang and Liang 2019) add Gaussian noise to the input and then quantize it. They report that this defense outperforms all of the NeurIPS 2017 submissions. For their attack, Zhang et al. approximate the gradient of the transform, which could lead to a sub-optimal attack. In this paper, we use the exact gradients for all transformations when available.

More recently, Raff et al. (Raff et al. 2019) claim to achieve a state-of-the-art robust accuracy $24\times$ better than adversarial training using a random transformation defense known as Barrage of Random Transforms (BaRT). BaRT involves randomly sampling a large set of image transformations and applying them to the input in a random order. Because many transformations are non-differentiable, BaRT evaluates their scheme using an attack that approximates the gradients of the transforms. In Section 4, we show that this approximation is ineffective, giving overly optimistic impression of BaRT's robustness, and we re-evaluate BaRT using a stronger attack which utilizes exact transform gradients.

## 3 Random Transformation Defense

Here, we introduce notations and the design of our RT defense, formalizing the BaRT defense.

### 3.1 Decision Rules

RT repeatedly applies a randomly chosen transform to the input, uses a neural network to make a prediction, and then averages the softmax prediction scores:

$$g(x) := \mathbb{E}_{\theta \sim p(\theta)} \left[ \sigma \left( f \left( t(x; \theta) \right) \right) \right] \qquad (2)$$

where $\sigma(\cdot)$ is the softmax function, $f : \mathbb{R}^d \to \mathbb{R}^C$ a neural network ($C$ is the number of classes), and the transformation $t(\cdot; \theta) : \mathbb{R}^d \to \mathbb{R}^d$ is parameterized by a random variable $\theta$ drawn from some distribution $p(\theta)$.

In practice, we approximate the expectation in Eqn. 2 with

$n$ Monte Carlo samples per one input $x$:

$$g(x) \approx g_n(x) := \frac{1}{n} \sum_{i=1}^{n} \sigma\left(f(t(x; \theta_i))\right) \qquad (3)$$

We then define the final prediction as the class with the largest softmax probability: $\hat{y}(x) = \arg\max_{c \in [C]} [g_n(x)]_c$. Note that this decision rule is different from most previous works that use a majority vote on hard labels, i.e., $\hat{y}_{\mathrm{maj}}(x) = \arg\max_{c \in [C]} \sum_{i=1}^{n} \mathbb{1}\left\{c = \arg\max_{j \in [C]} f_j(x)\right\}$ (Raff et al. 2019; Cohen, Rosenfeld, and Kolter 2019). We later show in Appendix D.1 that our rule is empirically superior to the majority vote. From the Law of Large Numbers, as $n$ increases, the approximation in Eqn. 3 converges to the expectation in Eqn. 2. Fig. 1 illustrates the structure and the components of the RT architecture.

## 3.2 Parameterization of Transformations

Here, $t(\cdot; \theta)$ represents a composition of $S$ different image transformations where $\theta = \{\theta^{(1)}, \ldots, \theta^{(S)}\}$ and $\theta^{(s)}$ denotes the parameters for the $s$-th transformation, i.e.,

$$t(x; \theta) = t_{\theta^{(S)}} \circ t_{\theta^{(S-1)}} \circ \cdots \circ t_{\theta^{(1)}}(x) \qquad (4)$$

Each $\theta^{(s)}$ is a random variable comprised of three components, i.e., $\theta^{(s)} = \{\tau^{(s)}, \beta^{(s)}, \alpha^{(s)}\}$, which dictate the properties of a transformation:

1. *Type* $\tau$ of transformation to apply (e.g., rotation, JPEG compression), which is uniformly drawn, without replacement, from a pool of $K$ transformation types: $\tau \sim \mathrm{Cat}(K, \mathbf{1}/K)$.
2. A *boolean* $\beta$ indicating whether the transformation will be applied. This is a Bernoulli random variable with probability $p_\beta$: $\beta \sim \mathrm{Bern}(p)$.
3. *Strength* of the transformation (e.g., rotation angle, JPEG quality) denoted by $\alpha$, sampled from a predefined distribution (either uniform or normal): $\alpha \sim p(a)$.

Specifically, for each of the $n$ transformed samples, we sample a permutation of size $S$ out of $K$ transformation types in total, i.e. $\{\tau^{(1)}, \ldots, \tau^{(S)}\} \in \mathrm{Perm}(K, S)$. Then the boolean and the strength of the $s$-th transform are sampled: $\beta^{(s)} \sim \mathrm{Bern}(p_{\tau^{(s)}})$ and $\alpha^{(s)} \sim p(a_{\tau^{(s)}})$. We abbreviate this sampling process as $\theta \sim p(\theta)$ which is repeated for every transformed sample (out of $n$) for a single input.

Assuming that the $K$ transformation types are fixed, an RT defense introduces, at most, $2K$ hyperparameters, $\{p_1, \ldots, p_K\}$ and $\{a_1, \ldots, a_K\}$, that can be tuned. It is also possible to tune by selecting $K'$ out of $K$ transformation types, but this is combinatorially large in $K$. In Appendix C, we show a heuristic for "pruning" the transformation types through tuning $p$ and $a$ (e.g., setting $p = 0$ is equivalent to removing that transformation type).

## 3.3 Choices of Transformations

In this work, we use a pool of $K = 33$ different image transformations including 19 differentiable and 2 non-differentiable transforms taken from the 30 BaRT transforms (Raff et al. 2019) (counting each type of noise injection

as its own transform). We replace non-differentiable transformations with a smooth differentiable alternative (Shin and Song 2017). The transformations fall into seven groups: noise injection (7), blur filtering (4), color-space alteration (8), edge detection (2), lossy compression (3), geometric transformation (5), and stylization (4). All transforms are described in Appendix A.1.

## 4 Evaluating Raff et al. (2019)'s BaRT

Backward-pass differentiable approximation (BPDA) was proposed as a heuristic for approximating gradients of non-differentiable components in many defenses to make gradient-based attacks applicable (Athalye, Carlini, and Wagner 2018). It works by first approximating the function with a neural network and backpropagate through this network instead of the non-differentiable function. Evaluations of BaRT in Raff et al. (2019) have considered BPDA as some transformations are innately non-differentiable or have zero gradients almost everywhere (e.g., JPEG compression, precision reduction, etc.). To approximate a transformation, we train a model $\tilde{t}_\phi$ that minimizes the Euclidean distance between the transformed image and the model output:

$$\min_\phi \sum_{i=1}^{N} \mathbb{E}_{\theta \sim p(\theta)} \left\| \tilde{t}_\phi(x_i; \theta) - t(x_i; \theta) \right\|_2 \qquad (5)$$

We evaluate the BPDA approximation below in a series of experiments that compare the effectiveness of the BPDA attack to an attack that uses exact gradients.

### 4.1 Experiment Setup

Our experiments use two datasets: CIFAR-10 and Imagenette (Howard 2021), a ten-class subset of ImageNet. While CIFAR-10 is the most common benchmark in the adversarial robustness domain, some image transformations work poorly on low-resolution images. We choose Imagenette because BaRT was created on ImageNet, but we do not have resources to do thorough investigation on top of adversarial training on ImageNet. Additionally, the large and realistic images from Imagenette more closely resemble real-world usage All Imagenette models are pre-trained on ImageNet to speed up training and boost performance. Since RT models are stochastic, we report their average accuracy together with the 95% confidence interval from 10 independent runs. Throughout this work, we consider the perturbation size $\epsilon$ of $16/255$ for Imagenette and $8/255$ for CIFAR-10. Appendix A.2 has more details on the experiments (network architecture, hyperparameters, etc.).

### 4.2 BPDA Attack is Not Sufficiently Strong

We re-implemented and trained a BaRT model on these datasets, and then evaluated the effectiveness of BPDA attacks against this model.[1] First, we evaluate the full BaRT model in Table 1, comparing an attack that uses a BPDA approximation (as Raff et al. (2019)) vs an attack that uses the exact gradient for differentiable transforms and

---

[1]The authors have been very helpful with the implementation details but cannot make the official code or model weights public.

| Transforms used | Clean accuracy | Adversarial accuracy w/ gradient approximations | | | |
|---|---|---|---|---|---|
| | | Exact | BPDA | Identity | Combo |
| BaRT (full) | $88.10 \pm 0.16$ | n/a | $52.32 \pm 0.22$ | $36.49 \pm 0.25$ | $\mathbf{25.24 \pm 0.16}$ |
| BaRT (only differentiable) | $87.43 \pm 0.28$ | $\mathbf{26.06 \pm 0.21}$ | $65.28 \pm 0.25$ | $41.25 \pm 0.26$ | n/a |

Table 1: Comparison of attacks with different gradient approximations. "Exact" directly uses the exact gradient. "BPDA" uses the BPDA gradient for most transforms and the identity for a few. "Identity" backpropagates as an identity function, and "Combo" uses exact gradient for differentiable transforms and BPDA gradient otherwise. Full BaRT uses a nearly complete set of BaRT transforms ($K = 26$), and "BaRT (only differentiable)" uses only differentiable transforms ($K = 21$). We use PGD attack with EoT and CE loss ($\epsilon = 16/255$, 40 steps).

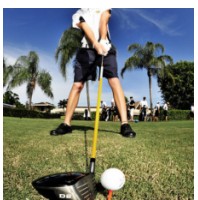 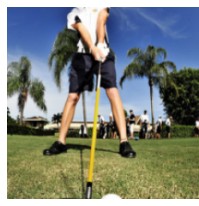 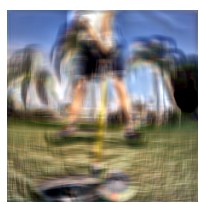

(a) Original      (b) Exact crop      (c) BPDA crop

Figure 2: Comparison of crop transform output and output of BPDA network trained to approximate crop transform.

BPDA for non-differentiable transforms, denoted "BPDA" and "Combo", respectively. Empirically, we observe that attacks using BPDA are far weaker than the equivalent attack using exact gradient approximations. Similarly, on a variant BaRT model that uses only the subset of differentiable transforms, the BDPA attack is worse than an attack that uses the exact gradient for all transforms. BPDA is surprisingly weaker than even a naive attack which approximates all transform gradients with the identity. There are a few possible explanations for the inability of BPDA to approximate transformation gradients well:

1. As Fig. 2 illustrates, BPDA struggles to approximate some transforms accurately. This might be partly because the architecture Raff et al. (2019) used (and we use) to approximate each transform has limited functional expressivity: it consists of five convolutional layers with 5x5 kernel and one with 3x3 kernel (all strides are 1), so a single output pixel can only depend on the input pixels fewer than 11 spaces away in any direction ($5 \cdot \lfloor \frac{5}{2} \rfloor + 1 \cdot \lfloor \frac{3}{2} \rfloor = 11$). Considering the inputs for Imagenette are of size $224 \times 224$, some transforms like "crop" which require moving pixels much longer distances are impossible to approximate with such an architecture.
2. The BPDA network training process for solving Eqn. 5 may only find a sub-optimal solution, yielding a poor approximation of the true transformation.
3. During the attack, the trained BPDA networks are given partially transformed images, yet the BPDA networks are only trained with untransformed inputs.
4. Since we are backpropagating through several transforms, one poor transform gradient approximation could ruin the overall gradient approximation.

Appendix A.3 has more details on these experiments. These results show that BaRT's evaluation using BPDA was overly optimistic, and BaRT is not as robust as previously thought.

Since BPDA is unreliable for approximating gradients of non-differentiable image transformations, **we recommend that other ensuing RT-based defenses only use differentiable transformations.** For the rest of this paper, we only study the robustness of RT defenses with differentiable transforms to isolate them from an orthogonal line of research on non-differentiable defenses (e.g., with approximate gradients or zero-th order attacks). Additionally, differentiable models can also boost their robustness further when combined with adversarial training. We explore this direction further in Section 7. Even without non-differentiable transforms, we still lack reliable evaluation on stochastic defenses apart from EoT. In the next section, we show that applying an EoT attack on RT defense results in a critically sub-optimal evaluation. After that, we propose a stronger attack.

## 5 Hyperparameter Tuning on RT Defenses

Before investigating attacks, we want to ensure we evaluate on the most robust RT defense possible. We found that BaRT is not robust, but it could be because of the chosen transformations and their hyperparameters which they do not provide any justification for. Finding the most robust RT defense is, however, challenging because it consists of numerous hyperparameters including the $K$ transformation types, the number of transformations to apply ($S$), and their parameters ($a$ and $p$). A typical grid search is intractable since we have 33 transformations, and trying to optimize the parameters directly with the reparameterization trick does not work as most transforms are not differentiable w.r.t. their parameters.

We systematically address this problem by using Bayesian optimization (BO) (Snoek, Larochelle, and Adams 2012), a well-known black-box optimization technique used for hyperparameter search, to fine-tune $a$ and $p$. In short, BO optimizes an objective function that takes in the hyperparameters ($a$ and $p$ in our case) as inputs and outputs adversarial accuracy. This process, which is equivalent to one iteration in BO, is computationally expensive as it involves training a neural network as a backbone for an RT defense and evaluating it with our new attack. Consequently, we have to scale down the problem by shortening the training, using fewer training/testing data samples, and evaluating with fewer attack steps. Essentially, we have to trade off precision of the search for efficiency. Because BO does not natively support categorical or integral

| Datasets | Attacks | Adv. Accuracy |
|---|---|---|
| Imagenette | Baseline | $70.79 \pm 0.53$ |
| | AutoAttack | $85.46 \pm 0.43$ |
| | Our attack | $\mathbf{6.34} \pm 0.35$ |
| CIFAR-10 | Baseline | $33.83 \pm 0.44$ |
| | AutoAttack | $61.13 \pm 0.85$ |
| | Our attack | $\mathbf{29.91} \pm 0.35$ |

Table 2: Comparison between the baseline EoT attack (Atha-lye et al. 2018), AutoAttack (Croce and Hein 2020), and our attack on the RT defense whose transformation parameters have been fine-tuned by Bayesian Optimization to maximize the robustness. For AutoAttack, we use its standard version combined with EoT. For Imagenette, we use $\epsilon = 16/255$, for CIFAR-10, $\epsilon = 8/255$.

---

**Algorithm 1:** Our best attack on RT defenses

**Input:** Set of $K$ transformations and distributions of their parameters $p(\theta)$, neural network $f$, perturbation size $\epsilon$, max. PGD steps $T$, step size $\{\gamma_t\}_{t=1}^T$, and AggMo's damping constants $\{\mu_b\}_{b=1}^B$.
**Output:** Adversarial examples $x_{\text{adv}}$
**Data:** Test input $x$ and its ground-truth label $y$
`// Initialize x_adv and velocities`
1 $x_{\text{adv}} \leftarrow x + u \sim \mathcal{U}[-\epsilon, \epsilon], \quad \{v_b\}_{b=1}^B \leftarrow \mathbf{0}$
2 **for** t $\leftarrow$ 1 **to** $T$ **do**
3     $\{\theta_i\}_{i=1}^n \sim p(\theta)$
    `// Compute a gradient estimate with`
    `   linear loss on logits`
    `   (Section 6.2) and with SGM`
    `   (Section 6.3)`
4     $G_n \leftarrow \nabla \mathcal{L}_{\text{Linear}} \left( \frac{1}{n} \sum_{i=1}^n f(t(x_{\text{adv}}; \theta_i)), y \right)$
5     $\hat{G}_n \leftarrow \text{sign}(G_n)$    `// Use signed gradients`
    `// Update velocities and x_adv with`
    `   AggMo (Section 6.4)`
6     **for** b $\leftarrow$ 1 **to** $B$ **do**
7        $v_b \leftarrow \mu_b \cdot v_b + \hat{G}_n$
8     $x_{\text{adv}} \leftarrow x_{\text{adv}} + \frac{\gamma_t}{B} \sum_{b=1}^B v_b$
9 **return** $x_{\text{adv}}$

---

variables, we experiment with different choices for $K$ and $S$ without the use of BO. The full details of this procedure are presented Appendix C.

# 6 State-of-the-Art Attack on RT Defenses

We propose a new attack on differentiable RT defenses that leverages insights from previous literature on transfer attacks as well as recent stochastic optimization algorithms. Our attack is immensely successful and shows that even the fine-tuned RT defense from Section 5 shows almost no adversarial robustness (Table 2). We summarize our attack in Algorithm 1 before describing the setup and investigating the three main design choices that make this attack successful and outperform the baseline from Athalye et al. (2018) by a large margin.

## 6.1 Setup: Stochastic Gradient Method

First, we describe the setup and explain intuitions around variance of the gradient estimates. Finding adversarial examples on RT defenses can be formulated as the following stochastic optimization problem:

$$\max_{\delta: \|\delta\|_\infty \leq \epsilon} H(\delta) \coloneqq \max_{\delta: \|\delta\|_\infty \leq \epsilon} \mathbb{E}_\theta \left[ h(\delta; \theta) \right] \quad (6)$$

$$\coloneqq \max_{\delta: \|\delta\|_\infty \leq \epsilon} \mathbb{E}_\theta \left[ \mathcal{L}(f(t(x + \delta; \theta)), y) \right] \quad (7)$$

for some objective function $\mathcal{L}$. Note that we drop dependence on $(x, y)$ to declutter the notation. Since it is not possible to evaluate the expectation or its gradients exactly, the gradients are estimated by sampling $\{\theta_i\}_{i=1}^n$ similarly to how we obtain a prediction $g_n$. Suppose that $H$ is smooth and convex, and variance of the gradient estimates is bounded by $\sigma^2$, i.e.,

$$\mathbb{E}_{\theta \sim p(\theta)} \left[ \|\nabla h(\delta; \theta) - \nabla H(\delta)\|^2 \right] \leq \sigma^2, \quad (8)$$

the error of SGD after $T$ iterations is $\mathcal{O}\left( 1/T + \sigma/\sqrt{T} \right)$ for an appropriate step size (Ghadimi and Lan 2013). This result suggests that small $\sigma$ or low-variance gradient speeds up convergence which is highly desirable for attackers and defenders alike. Specifically, it leads to more efficient and more accurate evaluation as well as a stronger attack to use during adversarial training, which in turn, could yield a better defense (we explore this in Section 7).

As a result, the analyses on our attack will be largely based on variance and two other measures of spread of the gradients. Specifically, we measure (1) the dimension-averaged variance in Eqn. 8, (2) cosine similarity and (3) a percentage of matching signs between mean gradient and each gradient sample. Since all three metrics appear to be highly correlated in theory and in practice, we only report the variance in the main paper. For the other metrics and their mathematical definitions, please see Appendix B.3.

**EoT Baseline.** We compare our attack to the baseline which is exactly taken from Athalye et al. (2018). This attack takes on the same form as Eqn. 7 and its gradients are averaged over $n$ gradient samples:

$$H_n^{\text{EoT}}(\delta) \coloneqq \frac{1}{n} \sum_{j=1}^n \mathcal{L}\left( f\left( t(x + \delta; \theta_j) \right), y \right) \quad (9)$$

It is important to note that this approximation does not exactly match the decision rule of RT defenses as the expectation should be in front of $f$ but behind the loss function (see Eqn. 2). While the gradient estimates from Eqn. 9 are unbiased, they may have high variance as each gradient sample is equivalent to computing the loss on $g_n$ with $n = 1$. In the next section, we will compare other options for objective functions and decision rules and show that there are better alternatives to the original EoT.

**Signed gradients.** All of the attacks used in this study including ours and the baseline use signs of gradients instead of the gradients themselves. This is a common practice for gradient-based $\ell_\infty$-attacks, and we have also empirically confirm that it leads to much stronger attacks. This is also the

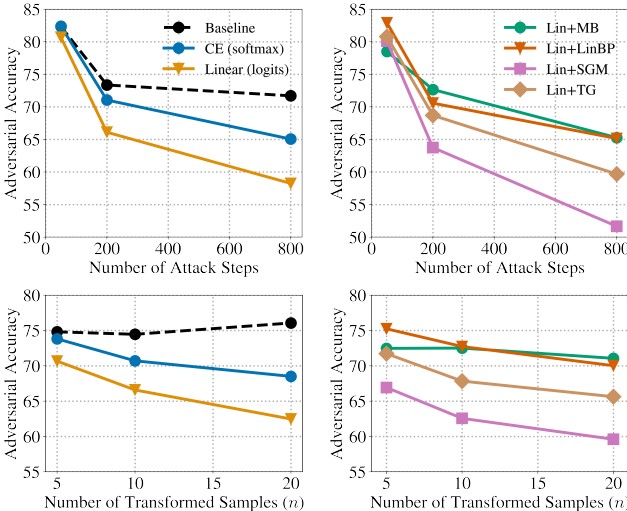

(a) Comparison among loss functions and decision rules

(b) Comparison among transfer attack techniques

Figure 3: Comparison of PGD attack's effectiveness with (a) different loss functions and decision rules, and (b) different attack variants with improved transferability. The error bars are too small to see with the markers so we report the numerical results in Table 4. "Baseline" refers to EoT with CE loss in Eqn. 9.

reason that we measure sign matching as a measure of spread of the gradient estimates. In addition to the $\ell_\infty$-constraint, using signed gradients as well as signed momentum is also beneficial as it has been shown to reduce variance for neural network training and achieve even faster convergence than normal SGD in certain cases (Bernstein et al. 2018).

## 6.2 Adversarial Objectives and Decision Rules

Here, we propose new decision rules and loss functions for the attacks as alternatives to EoT. Note that this need not be the same as the rule used for making prediction in Eqn. 2. First, we introduce *softmax* and *logits* rules:

$$H^{\text{softmax}}(\delta) \coloneqq \mathcal{L}\left(\mathop{\mathbb{E}}_{\theta \sim p(\theta)}\left[\sigma\left(f\left(t(x + \delta; \theta)\right)\right)\right], y\right) \quad (10)$$

$$H^{\text{logits}}(\delta) \coloneqq \mathcal{L}\left(\mathop{\mathbb{E}}_{\theta \sim p(\theta)}\left[f\left(t(x + \delta; \theta)\right)\right], y\right) \quad (11)$$

$H^{\text{softmax}}$, or loss of the expected softmax probability, is the same rule as the decision rule of RT defenses (Eqn. 2). It was also used by Salman et al. (2019) where $\mathcal{L}$ is cross-entropy loss. $H^{\text{logits}}$ or an expected logits, is similar to $H^{\text{softmax}}$ but without the softmax function to avoid potential vanishing gradients from softmax.

In addition to the rules, we experiment with two choices of $\mathcal{L}$ commonly used for generating adversarial examples: cross-entropy loss (CE) and linear loss (Linear). The linear loss is defined as the difference between the largest logit of

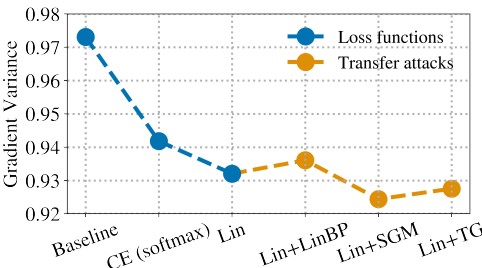

Figure 4: Comparison of dimension-normalized variance of the gradient estimates across (blue) different loss functions and decision rules and (yellow) transferability-improving attacks. Strong attacks are highly correlated with low variance of their gradient estimates, i.e., Lin+SGM. Note that Lin+MB or Momentum Boosting is not shown here because it does not modify the gradients.

the wrong class and logit of the correct class:

$$\mathcal{L}_{\text{Linear}}(x, y) \coloneqq \max_{j \neq y} F_j - F_y \quad (12)$$

$$\text{where} \quad F = \mathop{\mathbb{E}}_{\theta \sim p(\theta)}\left[f\left(t(x; \theta)\right)\right] \quad (13)$$

The advantage of the linear loss is that its gradient estimates are unbiased, similarly to EoT, meaning that the expectation can be moved in front of $\mathcal{L}$ due to linearity. However, this is not the case for CE loss.

**Attack evaluation and comparison.** We evaluate the attacks by their effectiveness in reducing the adversarial accuracy (lower means stronger attack) on the RT defense obtained from Section 5. In our setting, the adversarial examples are generated once, and then they are used to compute the accuracy 10 times, each with a different random seed on the RT defense. We report the average accuracy over these 10 runs together with the 95%-confidence interval. Alternatively, one can imagine a threat model that counts at least one misclassification among a certain number of trials as incorrect. This is an interesting and perhaps more realistic in some settings, but the optimal attack will be very different from EoT as we care a lot less about the expectation. This, however, is outside of the scope of our work.

In Fig. 3a, we compare the effectiveness of four attacks, each using a different pair of losses and decision rules with varying numbers of PGD steps and samples $n$. The widely used EoT method performs the worst of the four. CE loss on mean softmax probability performs better than EoT, confirming the observation made by Salman *et al.* (Salman et al. 2019). Linear loss and CE loss on average logits are even better and are consistently the strongest attacks, across all hyperparameters. For the rest of this paper, we adopt the linear loss with mean logits as the main objective function.

**Connection to variance.** As we predicted in Section 6.1, a stronger attack directly corresponds to lower variance. This hypothesis is confirmed by Fig. 4. For instance, the EoT baseline has the highest variance as well as the worst performance according to Fig. 5. On the other hand, the linear loss (Lin)

has the lowest variance among the three loss functions (blue) and hence, it performs the best. The other three points in orange will be covered in the next section.

## 6.3 Ensemble and Transfer Attacks

RT can be regarded as an ensemble with each member sharing the same neural network parameters but applying different sets of transformations to the input (i.e., different $\theta$'s from random sampling). Consequently, we may view a white-box attack on RT defenses as a "partial" black-box attack on an ensemble of (infinitely) many models where the adversary wishes to "transfer" adversarial examples generated on some subset of the members to another unseen subset.

Given this interpretation, we apply four techniques designed to enhance the transferability of adversarial examples to improve the attack success rate on RT defense. The techniques include momentum boosting (MB) (Dong et al. 2018), modifying backward passes by ignoring non-linear activation (LinBP) (Guo, Li, and Chen 2020) or by emphasizing the gradient through skip connections of ResNets more than through the residual block (SGM) (Wu et al. 2020), and simply using a targeted attack with the linear loss function (TG) (Zhao, Liu, and Larson 2021). In Fig. 3b, we compare these techniques combined with the best performing loss and decision rule from Section 6.2 (i.e., the linear loss on logits). Only SGM improves the attack success rate at all settings while the rest result in weaker attacks than the one without any of the techniques (denoted by "Linear (logits)" in Fig. 3a).

SGM essentially normalizes the gradients and scales ones from the residual blocks by some constant less than 1 (we use 0.5) to reduce its influence and prioritize the gradients from the skip connection. Wu et al. (2020) explain that SGM leads to better transferability because gradients through skip connections preserve "low-level information" which tends to transfer better. Intuitively, this agrees with our variance explanation as the increased transferability implies a stronger agreement among gradient samples and hence, less spread or lower variance.

## 6.4 Stochastic Optimization Algorithm

While most attacks on deterministic models can use naive PGD to solve Eqn. 1 effectively, this is not the case for stochastic models like the RT defense. Here, the adversary only has access to noisy estimates of the gradients, making it a strictly more difficult problem, and techniques used in the deterministic case may no longer apply.

As mentioned in Section 6.1, high-variance gradient estimates undermine the convergence rate of SGD. Thus, the attack should benefit from optimization techniques aimed at reducing the variance or speeding up the convergence of SGD. We first experiment with common optimizers such as SGD and Adam (Kingma and Ba 2015) with different hyperparameters, e.g., momentum, Nesterov acceleration, and learning rate schedules, to find the best setting for the linear loss with SGM. Based on this experiment, we found that a momentum term with an appropriate damping constant plays an important role in the attack success rate. Momentum is also well-known to accelerate and stabilize training of neural networks (Sutskever et al. 2013). Fig. 10a reports adversarial

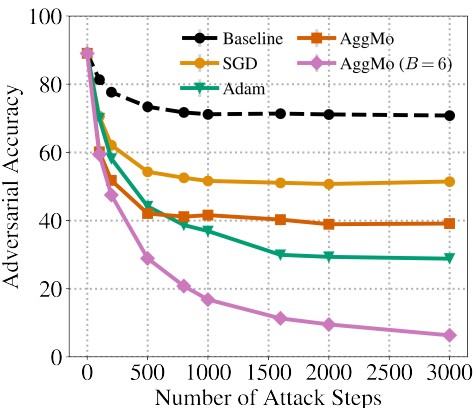

Figure 5: Comparison of the optimizers for attacking an RT defense with $\epsilon = 16/255, n = 10$ on Imagenette dataset. All but the baseline (CE loss with EoT) use the linear loss with SGM, and all but AggMo ($B = 6$) use the default hyperparameters. AggMo with $B = 6$ outperforms the other algorithms in terms of both the convergence rate and the final adversarial accuracy obtained. This result is not very sensitive to $B$ as any sufficiently large value ($\geq 4$) yields the same outcome.

accuracy at varying attack iterations and indicates that higher momentum constant leads to faster convergence and a higher attack success rate. However, the results seem highly sensitive to this momentum constant which also varies from one setting to another (e.g., number or types of transformations, dataset, etc.).

To mitigate this issue, we introduce another optimizer. AggMo is exactly designed to be less sensitive to choices of the damping coefficient by aggregating $B$ momentum terms with different constants instead of one (Lucas et al. 2019). After only a few tries, we found a wide range of values of $B$ where AggMo outperforms SGD with a fine-tuned momentum constant (see Fig. 10b). Fig. 5 compares the attacks using different choices of the optimizers to the baseline EoT attack. Here, the baseline can only reduce the adversarial accuracy from $89\%$ to $70\%$ while **our best attack manages to reach $6\%$ or over $4.3\times$ improvement.** This concludes that the optimizer plays a crucial role in the success of the attack, and **the RT defense, even with a carefully and systematically chosen transformation hyperparameters, is not robust against adversarial examples.**

Furthermore, we note that without our loss function and only using AggMo, the accuracy only goes down to $23\%$ at a much slower rate. Conversely, when the linear loss and SGM are used with SGD (no momentum), the accuracy drops to $51\%$. This signifies that all three techniques we deploy play important roles to the attack's effectiveness.

## 6.5 Comparison with AutoAttack

AutoAttack (Croce and Hein 2020) was proposed as a standardized benchmark for evaluating deterministic defenses against adversarial examples. It uses an ensemble of four different attacks that cover weaknesses of one another, one

| Defenses | Imagenette | | CIFAR-10 | |
|---|---|---|---|---|
| | Clean Accuracy | Adv. Accuracy | Clean Accuracy | Adv. Accuracy |
| Normal model | **95.41** | 0.00 | **95.10** | 0.00 |
| Madry et al. (2018) | 78.25 | **37.10** | 81.90 | 45.30 |
| Zhang et al. (2019) | 87.43 | 33.19 | 81.26 | **46.89** |
| RT defense | $89.04 \pm 0.34$ | $6.34 \pm 0.35$ | $81.12 \pm 0.54$ | $29.91 \pm 0.35$ |
| AdvRT defense | $88.83 \pm 0.26$ | $8.68 \pm 0.52$ | $80.69 \pm 0.66$ | $41.30 \pm 0.49$ |

Table 3: Comparison of RT and AdvRT defenses to prior robust deterministic models and a normally trained model. Both the RT and the AdvRT models on Imagenette lack the adversarial robustness. Conversely, the RT defense on CIFAR-10 does bring substantial robustness, and combining it with adversarial training boosts the adversarial accuracy further. Nonetheless, they still fall behind the previously proposed deterministic models including Madry et al. (2018) and Zhang et al. (2019). The largest number in each column is in bold.

of which does not use gradients. AutoAttack has been proven to be one of the strongest attack currently and is capable of catching defenses with false robustness caused by gradient obfuscation (Athalye, Carlini, and Wagner 2018).

While not particularly designed for stochastic models, AutoAttack can be used to evaluate them when combined with EoT. We report the accuracy on adversarial examples generated on AutoAttack with all default hyperparameters in the "standard" mode and 10-sample EoT in Table 2. AutoAttack performs worse than the baseline EoT and our attack on both Imagenette and CIFAR-10 by a large margin. One of the reasons is that AutoAttack is optimized for efficiency and so each of its attacks is usually terminated once a misclassification occurs. This is applicable to deterministic models, but for stochastic ones such as an RT defense, the adversary is better off finding the adversarial examples that maximize the expected loss instead of ones that are misclassified once.

To take this property into account, we include the accuracy reported by AutoAttack that treats a sample as incorrect if it is misclassified at least *once* throughout the entire process. For Imagenette, the accuracies after each of the four attacks (APGD-CE, APGD-T, FAB, and Square) is applied sequentially are 82.03, 78.81, 78.03, and 77.34, respectively. Note that this is a one-time evaluation so there is no error bar here. Needless to say, the adversarial accuracy computed this way is strictly lower than the one we reported in Table 2 and violates our threat model. However, it is still higher than that of the baseline EoT and our attack, suggesting that AutoAttack is ineffective against randomized models like RT defenses. AutoAttack also comes with a "random" mode for randomized models which only use APGD-CE and APGD-DLR with 20-sample EoT. The adversarial accuracies obtained from this mode are 85.62 and 83.83 or $88.62 \pm 0.46$ for single-pass evaluation as in Table 2. This random mode performs worse than the standard version.

## 7   Combining with Adversarial Training

To deepen our investigation, we explore the possibility of combining RT defense with adversarial training. However, this is a challenging problem on its own. For normal deterministic models, 10-step PGD is sufficient for reaching adversarial accuracy close to best known attack or the optimal adversarial accuracy. However, this is not the case for

RT defenses as even our new attack still requires more than one thousand iterations before the adversarial accuracy starts to plateau. Ultimately, the robustness of adversarially trained models largely depends on the strength of the attack used to generate the adversarial examples, and using a weak attack means that the obtained model will not be robust. A similar phenomenon is observed by Tramèr et al. (2018) and Wong, Rice, and Kolter (2020) where an adversarially trained model overfits to the weak FGSM attacks but has shown to be non-robust with the accurate evaluation. To test this hypothesis, we adversarially train the RT defense from Section 5 using our new attack with 50 iterations (already $5\times$ the common number of steps) and call this defense "AdvRT." The attack step size is also adjusted accordingly to $\epsilon/8$.

In Table 3, we confirm that training AdvRT this way results in a model with virtually no robustness improvement over the normal RT on Imagenette. On the other hand, the AdvRT trained on CIFAR-10 proves to be more promising even though it is still not as robust as deterministic models trained with adversarial training or TRADES (Zhang et al. 2019). Based on this result, **we conclude that a stronger attack on RT defenses that converge within a much fewer iterations will be necessary to make adversarial training successful.** In theory, it might be possible to achieve a robust RT model with 1,000-step attack on Imagenette, but this is too computationally intensive for us to verify, and it will not to scale to any realistic setting.

## 8   Conclusion

While recent papers report state-of-the-art robustness with RT defenses, our evaluations show that RT generally underperforms existing defenses like adversarial training when met with a stronger attack, even after fine-tuning the hyperparameters of the defense. Through our experiments, we found that non-differentiability and high-variance gradients can seriously inhibit adversarial optimization, so we recommend using only differentiable transformations along with their exact gradients in the evaluation of future RT defenses. In this setting, we propose a new state-of-the-art attack that improves significantly over the baseline (PGD with EoT) and show that RT defenses as well as their adversarially trained counterparts are not as robust to adversarial examples as they were previously believed to be.

# A  Experiment Details

## A.1  Details on the Image Transformations

The exact implementation of RT models and all the transformations will be released. Here, we provide some details on each of the transformation types and groups. Then, we describe how we approximate some non-differentiable functions with differentiable ones.

### Noise injection

- **Erase:** Set the pixels in a box with random size and location to zero.
- **Gaussian noise:** Add Gaussian noise to each pixel.
- **Pepper:** Zero out pixels with some probability.
- **Poisson noise:** Add Poisson noise to each pixel.
- **Salt:** Set pixels to one with some probability.
- **Speckle noise:** Add speckle noise to each pixel.
- **Uniform noise:** Add uniform noise to each pixel.

### Blur filtering

- **Box blur:** Blur with randomly sized mean filter.
- **Gaussian blur:** Blur with randomly sized Gaussian filter with randomly chosen variance.
- **Median blur:** Blur with randomly sized median filter.
- **Motion blur:** Blur with kernel for random motion angle and direction.

### Color-space alteration

- **HSV:** Convert to HSV color-space, add uniform noise, then convert back.
- **LAB:** Convert to LAB color-space, add uniform noise, then convert back.
- **Gray scale mix:** Mix channels with random proportions.
- **Gray scale partial mix:** Mix channels with random proportions, then mix gray image with each channel with random proportions.
- **Two channel gray scale mix:** Mix two random channels with random proportions.
- **One channel partial gray:** Mix two random channels with random proportions, then mix gray image with other channel.
- **XYZ:** Convert to XYZ color-space, add uniform noise, then convert back.
- **YUV:** Convert to YUV color-space, add uniform noise, then convert back.

### Edge detection

- **Laplacian:** Apply Laplacian filter.
- **Sobel:** Apply the Sobel operator.

### Lossy compression

- **JPEG compression:** Compress image using JPEG to a random quality.
- **Color precision reduction:** Reduce color precision to a random number of bins.
- **FFT perturbation:** Perform FFT on image and remove each component with some probability.

### Geometric transforms

- **Affine:** Perform random affine transformation on image.
- **Crop:** Crop image randomly and resize to original shape.
- **Horizontal flip:** Flip image across the vertical.
- **Swirl:** Swirl the pixels of an image with random radius and strength.
- **Vertical flip:** Flip image across the horizontal.

### Stylization

- **Color jitter:** Randomly alter the brightness, contrast, and saturation.
- **Gamma:** Randomly alter gamma.
- **Sharpen:** Apply sharpness filter with random strength.
- **Solarize:** Solarize the image.

### Non-differentiable (for BPDA Tests Only)

- **Adaptive histogram:** Equalize histogram in patches of random kernel size.
- **Chambolle denoise:** Apply Chambolle's total variation denoising algorithm with random weight (can be implemented differentiably but was not due to time constraints).
- **Contrast stretching:** Pick a random minimum and maximum pixel value to rescale intensities (can be implemented differentiably but was not due to time constraints).
- **Histogram:** Equalize histogram using a random number of bins.

### Unused transforms from BaRT

- **Seam carving:** Algorithm used in Raff et al. (2019) has been patented and is no longer available for open-source use.
- **Wavelet denoising:** The implementation in Raff et al. (2019) is incomplete.
- **Salt & pepper:** We have already used salt and pepper noise separately.
- **Non-local means denoising:** The implementation of NL means denoising in Raff et al. (2019) is too slow.

## A.2  Experiment Details

All of the experiments are evaluated on 1000 randomly chosen test samples. Since we choose the default $n$ to be 20 for inference and 10 for the attacks, the experiments are at least 10 times more expensive than usual, and we cannot afford enough computation to run a large number of experiments on the entire test set. The networks used in this paper are ResNet-34 (He et al. 2016a) for Imagenette and Pre-activation ResNet-20 (He et al. 2016b) for CIFAR-10. In all of the experiments, we use a learning rate of 0.05, batch size of 128, and weight decay of 0.0005. We use cosine annealing schedule (Loshchilov and Hutter 2017) for the learning rate with a period of 10 epochs which also doubles after every period. All models are trained for 70 epochs, and we save the weights with the highest accuracy on the held-out validation data (which does not overlap with the training or test set). For adversarially trained RT defenses, the cosine annealing step is set to 10 and the training lasts for 70 epochs to reduce the computation. To help the training converge faster, we pre-train these RT models on clean data before turning on adversarial training as suggested by Gupta, Dube, and Verma (2020).

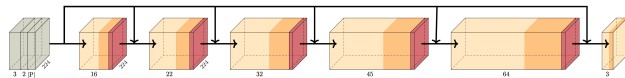

Figure 6: Fully-convolutional BPDA network from Raff et al. (2019). The network has six convolutional layers. All layers have a stride of 1. The first five layers have kernel size of 5 and padding size of 2, and the last layer has a kernel size of 3 and padding size of 1. The input consists of more than 5 channels, 3 of which are for the image RGB channels, 2 of which are CoordConv channels that include the coordinates of each pixel at that pixel's location, and the remaining channels are the parameters for the transformation copied at each pixel location. The network contains a skip connection from the input to each layer except the final layer.

### A.3 Details on BPDA Experiments

We used the following setup for the differentiability related experiments conducted in Section 4.2:

- Each accuracy is an average over 10 trials on the same set of 1000 Imagenette images.
- The defense samples $S = 10$ transforms from the full set of $K$ transforms.
- The image classifier uses a ResNet-50 architecture like in Raff et al. (2019) trained on transformed images for 30 epochs.
- The attack uses 40 PGD steps of size $4/255$ with an $\epsilon = 16/255$ to minimize the EoT objective.

The BPDA network architecture is the same used by Raff et al. (2019) and is outlined in Fig. 6. Here are more details on BPDA training:

- All BPDA networks were trained using Adam with a learning rate of 0.01 for 10 epochs.
- All networks achieve a per-pixel MSE below 0.01. The outputs of the BPDA networks are compared to the true transform outputs for several different transform types in Fig. 7.

The specific set of transforms used in each defense are the following:

- **BaRT (all):** adaptive histogram, histogram, bilateral blur, box blur, Gaussian blur, median blur, contrast stretching, FFT, gray scale mix, gray scale partial mix, two channel gray scale mix, one channel gray scale mix, HSV, LAB, XYZ, YUV, JPEG compression, Gaussian noise, Poisson noise, salt, pepper, color precision reduction, swirl, Chambolle denoising, crop.
- **BaRT (only differentiable):** all of the BaRT all transforms excluding adaptive histogram, histogram, contrast stretching, and Chambolle denoising.

# B  Details of the Attacks

## B.1 Differentiable Approximation

Some of the transformations contain non-differentiable operations which can be easily approximated with differentiable functions. Specifically, we approximate the rounding function in JPEG compression and color precision reduction, and the modulo operator in all transformations that require conversion between RGB and HSV color-spaces (HSV alteration and color jitter). Note that we are not using the non-differentiable transform on the forward pass and a differentiable approximation on the backward pass (like in BPDA). Instead, we are using the differentiable version both when performing the forward pass and when computing the gradient.

We take the approximation of the rounding function from Shin and Song (2017) shown in Eqn. 14.

$$\lfloor x \rceil_{\text{approx}} = \lfloor x \rceil + (x - \lfloor x \rceil)^3 \tag{14}$$

For the modulo or the remainder function, we approximate it using the above differentiable rounding function as a basis.

$$\text{mod}(x) = \begin{cases} x - \lfloor x \rceil & \text{if } x > \lfloor x \rceil \\ x - \lfloor x \rceil + 1 & \text{otherwise} \end{cases} \tag{15}$$

To obtain a differentiable approximation, we can replace the rounding operator with its smooth version in Eqn. 14. This function (approximately) returns decimal numbers or a fractional part of a given real number, and it can be scaled to approximate a modulo operator with any divisor.

Note that these operators are step functions and are differentiable almost everywhere, like ReLU. However, their derivatives are always zero (unlike ReLU), and so a first-order optimization algorithm would still fail on these functions.

## B.2 Effect of the Permutation of the Transformations

We mentioned in Section 3.2 that a permutation of the transforms $\{\tau^{(s)}\}_{s=1}^{S}$ is randomly sampled for each of the $n$ samples. However, we found that in practice, this leads to high-variance estimates of the gradients. On the other hand, fixing the permutation across $n$ samples in each attack iteration (i.e., $\tau$ is fixed but not $\alpha$ or $\beta$) results in lower variance and hence, a stronger attack, even though the gradient estimates are biased as $\tau$ is fixed. For instance, with fixed permutation, adversarial accuracy achieved by EoT attack is $51.44$ where the baseline EoT with completely random permutation is $70.79$. The variance also reduces from $0.97$ to $0.94$.

Additionally, the fixed permutation reduces the computation time as all transformations can be applied in batch. All of the attacks reported in this paper, apart from the baseline, use this fixed permutation.

## B.3 Variance of Gradients

We have described how we compute the sample variance of the gradients in Section 6.1. Here, we provide detailed calculations of the other three metrics. First, the unbiased variance is computed as normal with an additional normalization by dimension.

$$\mu_n := \frac{1}{n} \sum_{j=1}^{n} \nabla \hat{G}_{1,j} \tag{16}$$

$$\sigma_n^2 := \frac{1}{d} \frac{1}{n-1} \sum_{j=1}^{n} \left\| \mu_n - \hat{G}_{1,j} \right\|_2^2 \tag{17}$$

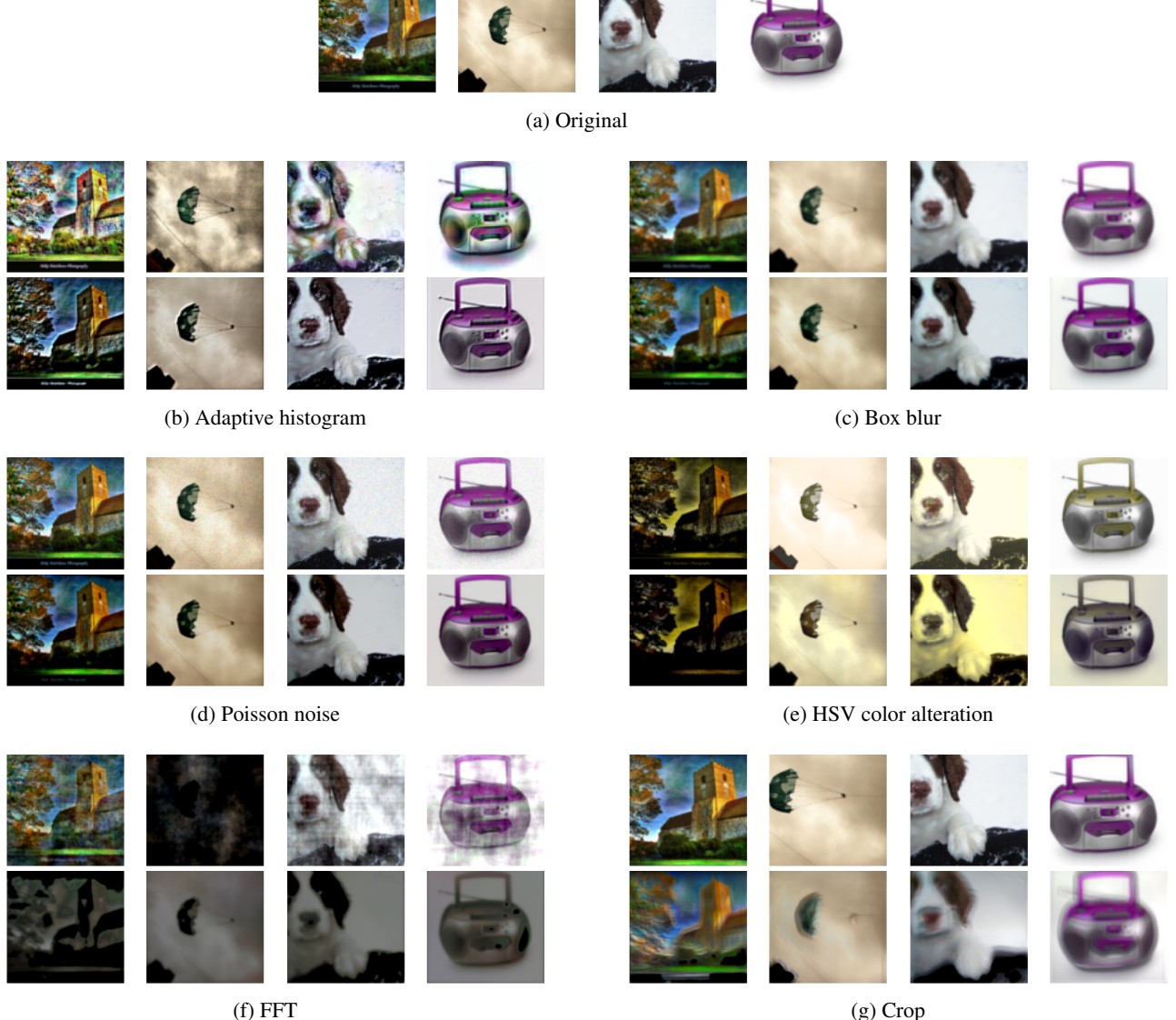

(a) Original

(b) Adaptive histogram

(c) Box blur

(d) Poisson noise

(e) HSV color alteration

(f) FFT

(g) Crop

Figure 7: Comparison of the true transformed outputs (top row) and outputs of respective BPDA networks (bottom row) for six different transformation types.

| Attacks | Adv. acc. with varying attack steps ($n=10$) | | | Adv. acc. with varying $n$ (attack steps = 200) | | |
|---|---|---|---|---|---|---|
| | 50 | 200 | 800 | 5 | 10 | 20 |
| Baseline | $82.34 \pm 0.43$ | $73.36 \pm 0.37$ | $71.70 \pm 0.39$ | $74.81 \pm 0.47$ | $74.46 \pm 0.55$ | $76.06 \pm 0.29$ |
| CE (softmax) | $82.37 \pm 0.39$ | $71.05 \pm 0.36$ | $65.06 \pm 0.39$ | $73.82 \pm 0.35$ | $70.71 \pm 0.53$ | $68.51 \pm 0.33$ |
| Linear (logits) | $80.67 \pm 0.50$ | $66.11 \pm 0.58$ | $58.26 \pm 0.62$ | $70.67 \pm 0.41$ | $66.59 \pm 0.57$ | $62.48 \pm 0.41$ |
| Linear+MB | $\mathbf{78.51} \pm 0.45$ | $72.66 \pm 0.50$ | $65.28 \pm 0.41$ | $72.47 \pm 0.39$ | $72.51 \pm 0.55$ | $71.06 \pm 0.32$ |
| Linear+LinBP | $82.90 \pm 0.50$ | $70.57 \pm 0.32$ | $65.15 \pm 0.43$ | $75.24 \pm 0.35$ | $72.73 \pm 0.40$ | $70.02 \pm 0.31$ |
| Linear+SGM | $80.10 \pm 0.43$ | $\mathbf{63.75} \pm 0.21$ | $\mathbf{51.68} \pm 0.35$ | $\mathbf{66.93} \pm 0.43$ | $\mathbf{62.57} \pm 0.31$ | $59.61 \pm 0.55$ |
| Linear+TG | $80.78 \pm 0.56$ | $68.70 \pm 0.34$ | $\mathbf{59.69} \pm 0.57$ | $71.72 \pm 0.41$ | $67.84 \pm 0.50$ | $65.63 \pm 0.50$ |

Table 4: Comparison of different attack techniques on our best RT model. Lower means stronger attack. This table only shows the numerical results plotted in Fig. 3.

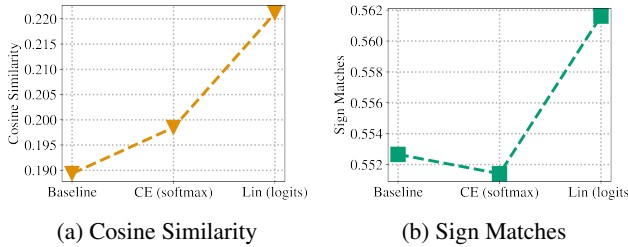

(a) Cosine Similarity (b) Sign Matches

Figure 8: (a) Cosine similarity and (b) percentage of sign matches for three pairs of attack loss functions and decision rules: CE loss with EoT "Baseline", CE loss on mean softmax probability "CE (softmax)", and linear loss on logits "Lin (logits)".

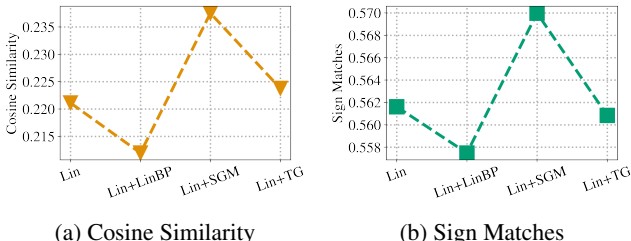

(a) Cosine Similarity (b) Sign Matches

Figure 9: (a) Cosine similarity and (b) percentage of sign matches for the linear loss and its combinations with three transfer attack techniques: Linear Backward Pass "LinBP", Skip Gradient Method "SGM", and targeted "TG".

where $\hat{G}_1$ is the signed gradients where the loss is estimated with one sample as defined in Algorithm 1.

The cosine similarity is computed between the mean gradient and all $n$ samples and then averaged.

$$\cos_n \coloneqq \frac{1}{n} \sum_{j=1}^{n} \frac{\left\langle \hat{G}_{1,j}, \mu_n \right\rangle}{\left\| \hat{G}_{1,j} \right\|_2 \cdot \|\mu_n\|_2} \quad (18)$$

Lastly, the sign matching percentage is

$$\text{sign\_match}_n. \coloneqq \frac{1}{n} \sum_{j=1}^{n} \frac{1}{d} \sum_{i=1}^{d} \mathbb{1}\{[\hat{G}_{1,j}]_i = [\mu_n]_i\} \quad (19)$$

Fig. 8 and Fig. 9 plot the cosine similarly and the sign matching for varying loss functions and varying transfer attacks, respectively. Similarly to Fig. 4, better attacks result in less spread of the gradient samples which corresponds to higher cosine similarity and sign matching percentage.

## C Details on Bayesian Optimization

One major challenge in implementing an RT defense is selecting the defense hyperparameters which include the $K$ transformation types, the number of transformations to apply ($S$), and their parameters ($a$ and $p$). To improve the robustness of RT defense, we use Bayesian optimization (BO), a well-known black-box optimization technique, to fine-tune $a$ and $p$ (Snoek, Larochelle, and Adams 2012). In this case, BO models the hyperparameter tuning as a Gaussian process where the objective function takes in $a$ and $p$, trains a neural network as a backbone for an RT defense, and outputs adversarial accuracy under some pre-defined $\ell_\infty$-budget $\epsilon$ as the metric used for optimization.

Since BO quickly becomes ineffective as we increase the dimensions of the search space, we choose to tune either $a$ or $p$, never both, for each of the $K$ transformation types. For transformations that have a tunable $a$, we fix $p = 1$ (e.g., noise injection, affine transform). For the transformations without an adjustable strength $a$, we only tune $p$ (e.g., Laplacian filter, horizontal flip). Additionally, because BO does not natively support categorical or integral variables, we experiment with different choices for $K$ and $S$ without the use of BO. Therefore, our BO problem must optimize over $K$ (up to 33) variables, far more than are typically present when doing model hyperparamter tuning using BO.

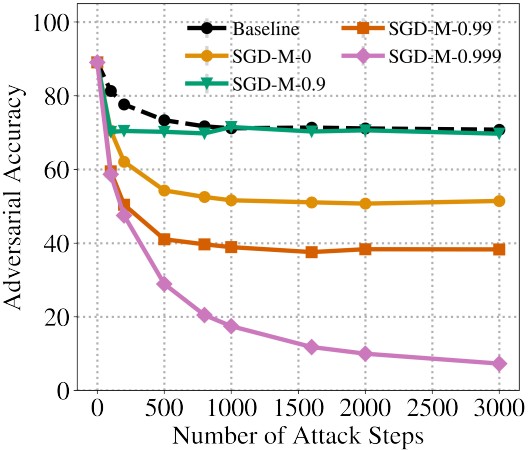

(a) SGD with varying momentum constants

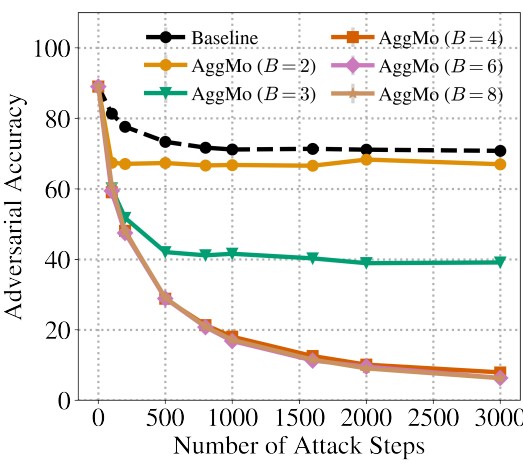

(b) AggMo with varying $B$'s

Figure 10: Effectiveness of the optimizers, (a) SGD and (b) AggMo, with varying momentum parameters. Increasing $B$ for AggMo in this case monotonically reduces the final adversarial accuracy until $B = 4$ where it plateaus. This is more predictable and stable than increasing the momentum constant in SGD.

Mathematically, the objective function $\psi$ is defined as

$$\psi : [0, 1]^K \to \mathcal{R}_{\infty, \epsilon} \in [0, 1] \qquad (20)$$

where the input is $K$ real numbers between 0 and 1, and $\mathcal{R}_{\infty, \epsilon}$ denotes the adversarial accuracy or the accuracy on $x_{\text{adv}}$ as defined in Eqn. 1. Since $\psi$ is very expensive to evaluate as it involves training and testing a large neural network, we employ the following strategies to reduce the computation: (1) only a subset of the training and validation set is used, (2) the network is trained for fewer epochs with a cosine annealing learning rate schedule to speed up convergence (Loshchilov and Hutter 2017), and (3) the attack used for computing $\mathcal{R}_{\infty, \epsilon}$ is weaker but faster. Even with these speedups, one BO run still takes approximately two days to complete on two GPUs (Nvidia GeForce GTX 1080 Ti). We also experimented with

---

**Algorithm 2:** Tuning and training RT defense.

**Input:** Set of transformation types, $n$, $p$, $\epsilon$
**Output:** $g^*(\cdot), \mathcal{R}, \mathcal{R}_{p,\epsilon}$
**Data:** Training data $\left( \boldsymbol{X}^{\text{train}}, \boldsymbol{Y}^{\text{train}} \right)$, test data $\left( \boldsymbol{X}^{\text{test}}, \boldsymbol{Y}^{\text{test}} \right)$

// Starting Bayesian optimization (BO)

1  Sub-sample $\left( \boldsymbol{X}^{\text{train}}, \boldsymbol{Y}^{\text{train}} \right)$ and split it into BO's training data $\left( \boldsymbol{X}_{\text{BO}}^{\text{train}}, \boldsymbol{Y}_{\text{BO}}^{\text{train}} \right)$ and validation data $\left( \boldsymbol{X}_{\text{BO}}^{\text{val}}, \boldsymbol{Y}_{\text{BO}}^{\text{val}} \right)$.

2  $\mathcal{R}_{p,\epsilon}^* \leftarrow 0$        // Best adversarial accuracy

3  $\{(p_i^*, \alpha_i^*)\}_{i=1}^K \leftarrow 0$               // Best RT hyperparameters

4  **for** step $\leftarrow 0$ **to** *MAX_BO_STEPS* **do**
        // Running one trial of BO
5  $\quad$ BO specifies $\{(p_i, \alpha_i)\}_{i=1}^K$ to evaluate.
6  $\quad$ Train an RT model on $\left( \boldsymbol{X}_{\text{BO}}^{\text{train}}, \boldsymbol{Y}_{\text{BO}}^{\text{train}} \right)$ with hyperparameters $\{(p_i, \alpha_i)\}_{i=1}^K$ to obtain $g$.
7  $\quad$ Test $g$ by computing $\mathcal{R}_{p,\epsilon}$ on $\left( \boldsymbol{X}_{\text{BO}}^{\text{val}}, \boldsymbol{Y}_{\text{BO}}^{\text{val}} \right)$ using a weak but fast attack.
8  $\quad$ **if** $\mathcal{R}_{p,\epsilon} > \mathcal{R}_{p,\epsilon}^*$ **then**
9  $\quad\quad$ $\mathcal{R}_{p,\epsilon}^* \leftarrow \mathcal{R}_{p,\epsilon}$
10 $\quad\quad$ $\{(p_i^*, \alpha_i^*)\}_{i=1}^K \leftarrow \{(p_i, \alpha_i)\}_{i=1}^K$
11 $\quad$ **else if** *No improvement for some steps* **then**
12 $\quad\quad$ break;

// Full training of RT

13 Train an RT model on $\left( \boldsymbol{X}^{\text{train}}, \boldsymbol{Y}^{\text{train}} \right)$ with best hyperparameters $\{(p_i^*, \alpha_i^*)\}_{i=1}^K$ to obtain $g^*$.
14 Evaluate $g^*$ by computing $\mathcal{R}$ and $\mathcal{R}_{p,\epsilon}$ on $\left( \boldsymbol{X}^{\text{test}}, \boldsymbol{Y}^{\text{test}} \right)$ using a strong attack.

---

other sophisticated hyperparameter-tuning algorithms based on Gaussian processes (Bergstra, Yamins, and Cox 2013; Kandasamy et al. 2020; Falkner, Klein, and Hutter 2018) but do not find them more effective. We summarize the main steps for tuning and training an RT defense in Algorithm 2.

We use the Ray Tune library for RT's hyperparameter tuning in Python (Liaw et al. 2018). The Bayesian optimization tool is implemented by Nogueira (2014), following analyses and instructions by Snoek, Larochelle, and Adams (2012) and Brochu, Cora, and de Freitas (2010). As mentioned in Section 5, we sub-sample the data to reduce computation for each BO trial. Specifically, we use 20% and 10% of the training samples for Imagenette and CIFAR-10 respectively (Algorithm 2, line 1) as Imagenette has a much smaller number of samples in total. The models are trained with the same transformations and hyperparameters used during inference, and here, $n$ is set to 1 during training, just as is done during standard data augmentation. We use 200 samples to evaluate each BO run in line 7 of Algorithm 2 with only 100 steps and $n = 10$.

One BO experiment executes two BO's in parallel. The maximum number of BO runs is 160, but we terminate the experiment if no improvement has been made in the last 40 runs after a minimum of 80 runs have taken place. The runtime depends on $S$ and the transformation types used. In our typical case, when all 33 transformation types are used and $S = 14$, one BO run takes almost an hour on an Nvidia

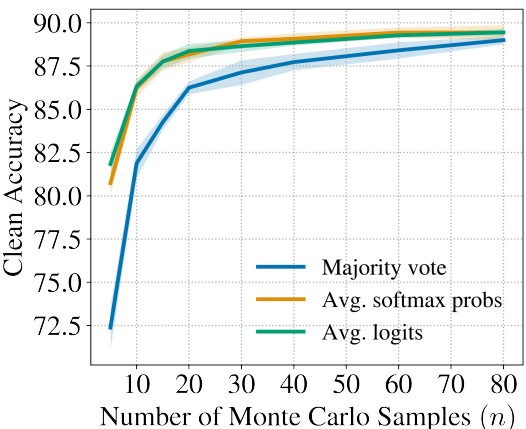

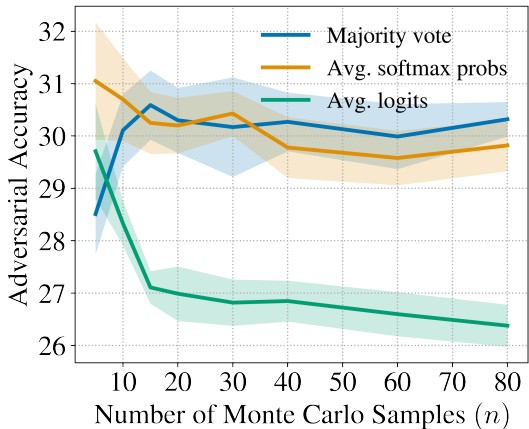

Figure 11: Clean accuracy of our best RT model computed with three decision rules for obtaining the final prediction from the $n$ output samples. The rules are majority vote (red), average softmax probability (blue), and average logits (green). The shaded areas represent the 95% confidence interval for each decision rule.

Figure 12: Adversarial accuracy ($\epsilon = 16/255$) of our best RT model computed with three decision rules for obtaining the final prediction from the $n$ output samples. The rules are majority vote (red), average softmax probability (blue), and average logits (green). The shaded areas represent the 95% confidence interval for each decision rule.

GeForce GTX 1080 Ti for Imagenette. One BO experiment then takes about two days to finish.

In line 13 and 14 of Algorithm 2, we now use the full training set and 1000 test samples as mentioned earlier. During the full training, $n$ is set to four which increases the training time by approximately four times. We find that using a larger $n$ is beneficial to both the clean and the adversarial accuracy, but $n$ larger than four does not make any significant difference.

### C.1 Details on the Final RT Model

We run multiple BO experiments (Algorithm 2) on different subsets of transformation types to identify which transformations are most/least effective in order to reduce $K$ as well as the number of hyperparameters our final run of BO has to tune. We then repeat Algorithm 2 initialized with the input-output pairs from the prior runs of BO to obtain a new set of hyperparameters. Finally, we remove the transformations whose $p$ or $a$ has been set to zero by the first run of BO, and we run BO once more with this filtered subset of transformations. At the end of this expensive procedure, we obtain the best and final RT model that we use in the experiments throughout this paper. For Imagenette, the final set of 18 transformation types used in this model are color jitter, erase, gamma, affine, horizontal flip, vertical flip, Laplacian filter, Sobel filter, Gaussian blur, median blur, motion blur, Poisson noise, FFT, JPEG compression, color precision reduction, salt noise, sharpen, and solarize. $S$ is set to 14.

## D Additional Experiments on the RT Model

### D.1 Decision Rules and Number of Samples

Fig. 11 and Fig. 12 compare three different decision rules that aggregate the $n$ outputs of the RT model to produce the final prediction $\hat{y}(x)$ given an input $x$. We choose the average softmax probability rule for all of our RT models because it

Table 5: RT's performance when only one of the transformation groups is applied. The attack is Linear+Adam+SGM with 200 steps and $n = 20$.

| Used Transformations | Clean Acc. | Adv. Acc. |
|---|---|---|
| Noise injection | $80.93 \pm 0.44$ | $\mathbf{8.35 \pm 0.20}$ |
| Blur filter | $97.32 \pm 0.20$ | $0.00 \pm 0.00$ |
| Color space | $94.40 \pm 0.53$ | $0.00 \pm 0.00$ |
| Edge detection | $97.64 \pm 0.09$ | $0.00 \pm 0.00$ |
| Lossy compression | $83.56 \pm 0.66$ | $3.56 \pm 0.26$ |
| Geometric transforms | $88.42 \pm 0.28$ | $0.83 \pm 0.21$ |
| Stylization | $\mathbf{98.31 \pm 0.09}$ | $0.00 \pm 0.00$ |

provides a good trade-off between the clean accuracy and the robustness. Majority vote has poor clean accuracy, and the average logits have poor robustness.

### D.2 Importance of the Transformation Groups

Choosing the best set of transformation types to use is a computationally expensive problem. There are many more transformations that can be applied outside of the 33 types we choose, and the number of possible combinations grows exponentially. BO gives us an approximate solution but is by no means perfect. Here, we take a step further to understand the importance of each transformation group. Table 5 gives an alternative way to gauge the contribution of each transformation group. According to this experiment, noise injection appears most robust followed by lossy compression and geometric transformations. However, this result is not very informative as most of the groups have zero adversarial accuracy, and the rest are likely to also reduce to zero given more attack steps. This result also surprisingly follows the commonly observed robustness-accuracy trade-off (Tsipras et al. 2019).

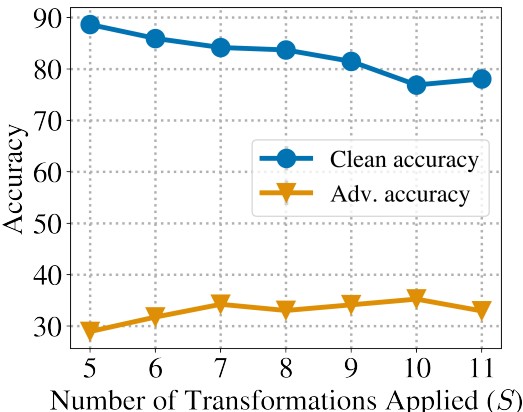

Figure 13: Adversarial accuracy of RT models obtained after running Algorithm 2 for different values of $S$ on CIFAR-10

### D.3 Number of Transformations

We test the effect of the transform permutation size $S$ on the clean and the robust accuracy of RT models (Fig. 13). We run Bayesian optimization experiments for different values of $S$ using all 33 transformation types, and all of the models are trained using the same procedure. Fig. 13 shows that generally more transformations (larger $S$) increase robustness but lower accuracy on benign samples.

## Acknowledgements

The first author of this paper was supported by the Hewlett Foundation through the Center for Long-Term Cybersecurity (CLTC), by the Berkeley Deep Drive project, by the National Science Foundation under Award CCF-1909204, and by generous gifts from Open Philanthropy and Google Cloud Research Credits program under Award GCP19980904.

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
