# OpenReview forum: "Demystifying the Adversarial Robustness of Random Transformation Defenses"
_AAAI.org/2022/Workshop/AdvML — AAAI-22 AdvML Workshop Oral_

### Official Review · Reviewer_vUox · 2021-11-29
**A more reliable evaluation of robustness for random transformation defenses**

**Rating:** 7
**Confidence:** 5

**Review:**

The authors show that backward-pass differentiable approximation struggles to approximate some transformations accurately, and non-differentiable transformation can’t guarantee robustness. Furthermore, the authors propose a novel attack method to evaluate random transformation defenses. This work provides a more reliable evaluation of robustness for random transformation defenses.

---

### Official Review · Reviewer_LPPi · 2021-11-30
**A novel method of empirical attacks and defenses using semantic transformations**

**Rating:** 8
**Confidence:** 4

**Review:**

This paper proposed novel defense and attack methods using image transformations. The random transformation defense is designed by averaging the scores of results under random transformations. This paper proposed to use Bayesian optimization to select the most informative transformation parameters. The authors also re-implemented and evaluate BaRT attacks and improve their performance.

The experimental results and solid and significant so I recommend a clear acceptance.

---

### Decision · Program_Chairs · 2021-12-01

**Decision:**

Accept (Oral)

**Comment:**

Both reviewers give high scores on this paper. Thus it is accepted as an oral presentation.